# EFFECTIVELY CLARIFY CONFUSION VIA VISUALIZED AGGREGATION AND SEPARATION OF DEEP REPRESENTATION

## ABSTRACT

Clarifying confusion is the most critical issue for improving classification performance. The current mainstream research mainly focuses on solving the confusion in a specific case, such as data insufficiency and class imbalance. In this paper, we propose a novel, simple and intuitive Aggregation Separation Loss (ASLoss), as an adjunct for classification loss to clarify the confusion in some common cases. The ASLoss aggregates the representations of the same class samples as near as possible and separates the representations of different classes as far as possible. We use two image classification tasks with three simultaneous confounding characteristics i.e. data insufficiency, class imbalance, and unclear class evidence to demonstrate ASLoss. Representation visualization, confusion comparison and detailed comparison experiments are conducted. The results show that representations in deep spaces extracted by ASLoss are sufficiently clear and distinguishable, the confusion among different classes is significantly clarified and the optimal network using ASLoss reaches the state-of-the-art level.

## 1 INTRODUCTION

Clarifying confusion is the most critical issue for improving classification performance. In fact, all prediction mistakes in classification are confusion, i.e., the model incorrectly considers samples of class "A" as class "B". Confusion occurs with almost all classification models but tends to be ignored in excellent-performing models because the mainstream datasets are artificially constructed to be nearly perfect Torralba et al. (2008); Russakovsky et al. (2015); Lin et al. (2014). However, the sample distribution in practical

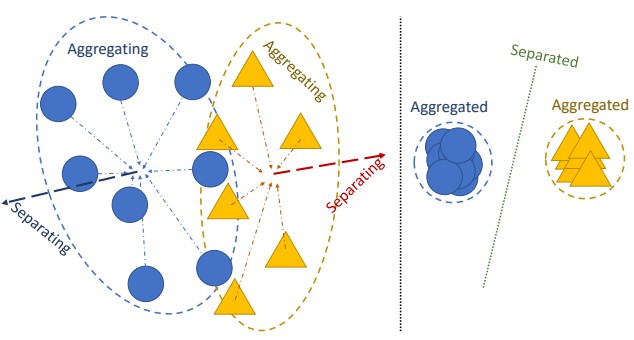

Figure 1: Schematic of Aggregation Separation Loss.

problems is often not as perfect as artificial datasets. First, collecting enough samples is difficult in some situations, such as the medical field due to privacy invasion Lin et al. (2021). High-performing models are extremely hard to obtain with insufficient data. Second, samples of different categories generally do not follow a uniform distribution but class imbalance Liu et al. (2019); Buda et al. (2018). The vast majority of samples belong to a small number of categories. Samples from unimpressive categories will be confused by the model into fat categories Cui et al. (2019). In addition, the classification boundaries of some problems are not clear Lin et al. (2022). Models will always confuse samples into similar classes. More seriously, many real-world problems simultaneously have more than one case confusing the models.

Many methods have been proposed to clarify confusion, but almost focus on insufficient data or class imbalance. For insufficient data, few-shot learning is a mainstream, including meta-learning Antoniou et al. (2019); Jamal et al. (2019); Lifchitz et al. (2019); Metz et al. (2019); Rajeswaran et al. (2019) and metric-based Zhang et al. (2020); Li et al. (2019); Sung et al. (2018) methods.

These methods have made great progress but the lack of data in few-shot learning is too extreme while its true generalization performance is still unclear. For class imbalance, resampling Chawla et al. (2002); Wang et al. (2019c); Zhou et al. (2020) and reweighting Cao et al. (2019); Cui et al. (2019) are two mainstreams. All these methods balance input sample frequency or loss of different classes but ignore the characteristic difference among categories.

In view of previous research, we believe that focusing on the class-level representation of samples **to mine the commonalities of the same class and the gaps among different classes** can clarify the confusion in these common fusion cases. As long as deep learning models can construct **sufficiently clear representations** during extracting features, the confusion under common conditions will be clarified. We propose an assistant to optimize the **representations into distinguishable** in the geometric space of deep features to efficiently clarify the confusion.

In this paper, we propose a novel, simple and intuitive method called Aggregation Separation Loss (ASLoss), as an adjunct for classification loss. This loss can be adopted on any linear feature extraction layers as shown in Figure 2, constructing distinguishable representations in geometric spaces of deep features to clarify the confusion in common cases as shown in 1. **It aggregates the representations of the same class samples as near as possible and separates the representations of different classes as far as possible to mine the commonalities of the same class and the gaps among different classes. To interpret its effect, the distinguishable representations can be visualized by condensing the representation layers into two dimensions.**

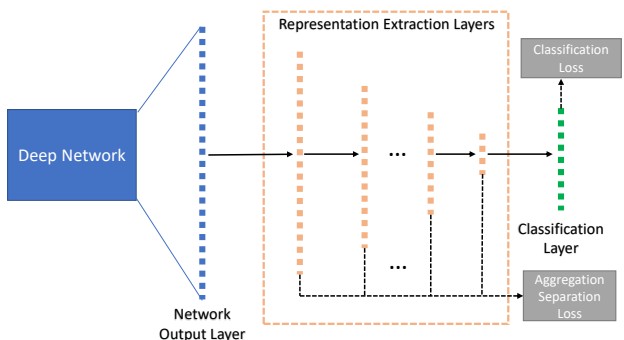

Figure 2: Schematic of Using Aggregation Separation Loss.

We validate our method using two image classification tasks that simultaneously have three easily confusion-caused common conditions: data insufficiency, class imbalance, and unclear evidence. The experimental results show that the representations in deep geometric spaces are sufficiently clearer, the performances of various deep networks are efficiently improved and the optimal network achieves state-of-the-art. The code for this work is available on GitHub[1].

## 2 RELATED WORK

**Contrastive learning** is a series of unsupervised methods that use an agent task to minimize the distances among varieties of the same sample and maximize the distances among different sample varieties Chen et al. (2020); Gao et al. (2021); Wang et al. (2021b); Bachman et al. (2019). These excellent methods using unsupervised pretrain improve downstream classification by finetuning. The key is using the agent task which transforms a sample into many varieties. Our method is plug-and-play without pretraining and transforming. And ASloss directly pulls and pushes the representations of samples but not varieties.

**Metric learning** also optimizes sample distances Wang et al. (2018b) Wang et al. (2018a; 2017a); Liu et al. (2017); Zhou et al. (2019); Wang et al. (2019a); Xu et al. (2019); Zheng et al. (2019); Wang et al. (2019b). Some of them propose better activation methods and others use linear transformation. Our method is similar to the linear transformation series but uses linear transformation on multiple layers, leading to more separable representations. Furthermore, our method does not set the class interval but sets a scope to make the distance between classes as large as possible, which is more straightforward.

**Triple loss** sets a triple of ($anchor, positive, negative$) to pull the same class samples and push other class samples Yuan et al. (2020); Schroff et al. (2015). ASLoss is more flexible, calculating all

---

[1]The link will be open if our work can be accepted.

samples directly. Similarly, ASLoss does not set the interval but sets the optimization scope so that classes are as far apart and evenly distributed as possible.

## 3 DEFINITION OF CONFUSION

For any sample $x \in \mathbf{X}$ where $\mathbf{X}$ is a category, if a person or a model considers $x$ to belong to another category $\mathbf{Y}$, this misperception is confusion. Confusion often occurs when the samples are very similar, the perceiver is inexperienced, and the class boundaries are not clear. For an example of similar samples, people often misidentify twins. For an example of inexperience, children who have never seen a tiger will consider a young tiger as a cat. The insufficient data and imbalanced class in machine learning belong to this case. For an example of the unclear boundary, it is difficult to determine whether a person is middle-aged or young based on only appearance.

## 4 AGGREGATION SEPARATION LOSS

The ASLoss is a simple and intuitive method including three parts: inner aggregation, outer separation and boundary constraint as shown in Figure 2.

### 4.1 INNER AGGREGATION

To mine the commonalities of the same class, the inner aggregation narrowing deep representations of the same class as near as possible is computed as:

$$dis_{inner} = \frac{1}{N_{c_i=c_j}} \sum_{i=1}^{N} \sum_{j=i}^{N} \mathbb{I}(c_i = c_j) D(r_i, r_j) \tag{1}$$

$$N_{c_i=c_j} = \sum_{i=1}^{N} \sum_{j=i}^{N} \mathbb{I}(c_i = c_j) \tag{2}$$

where $N$ is the total number of samples, $c_i$ denotes the class of sample $i$, $N_{c_i=c_j}$ is the total number of $c_i = c_j$ and $r_i$ refers to the representation in a deep geometric space output from a certain linear layer of sample $i$. $\mathbb{I}(c_i = c_j)$ is an indicator function whose value is 1 when $c_i = c_j$, otherwise the value is 0. $D(r_i, r_j)$ is a distance metric of $r_i$ and $r_j$ such as Euclidean distance etc. $dis_{inner}$ represents the average intra-class distance of all samples. Minimizing $dis_{inner}$ will aggregate the representations of the same class together during optimizing networks.

### 4.2 OUTER SEPARATION

To mine the gaps among different classes, the outer separation pushing the representations of different classes away as far as possible is calculated as:

$$dis_{outer} = \frac{1}{N_{c_i \neq c_j}} \sum_{i=1}^{N} \sum_{j=i}^{N} \mathbb{I}(c_i \neq c_j) D(r_i, r_j) \tag{3}$$

$$N_{c_i \neq c_j} = \sum_{i=1}^{N} \sum_{j=i}^{N} \mathbb{I}(c_i \neq c_j) \tag{4}$$

where $N$ is the total number of samples, $c_i$ denotes the class of sample $i$, $N_{c_i \neq c_j}$ is the total number of $c_i \neq c_j$ and $r_i$ refers to the representation in a deep geometric space output from a certain linear layer of sample $i$. $\mathbb{I}(c_i \neq c_j)$ is an indicator function whose value is 1 when $c_i \neq c_j$, otherwise the value is 0. $D(r_i, r_j)$ is a distance metric of $r_i$ and $r_j$. $dis_{outer}$ represents the average inter-class distance of all samples. Maximizing $dis_{outer}$ will separate the representations of different classes farther during optimizing networks.

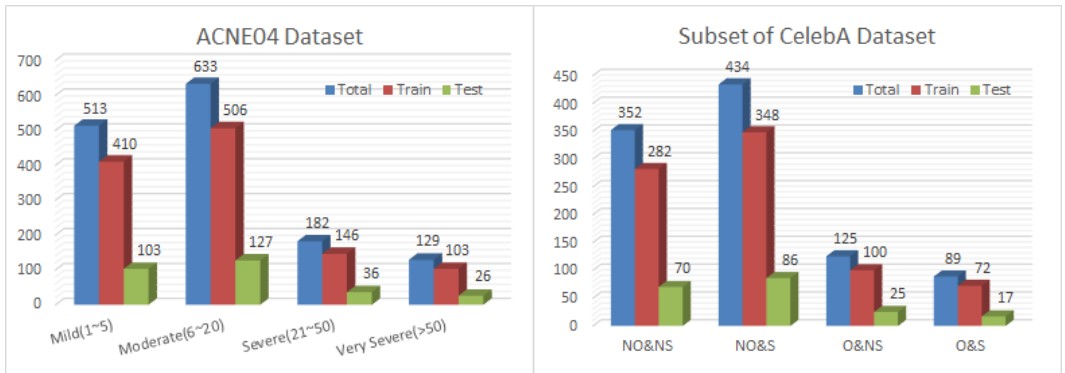

Figure 3: The total, trainset and testset sample numbers for each category of the ACNE04 and CelebA subset. (NO: mouth not open, NS: not smile, O: mouth open and S: smile)

### 4.3 BOUNDARY CONSTRAINT

Because the farthest distance in a space can be infinity, The optimization of equation 3 will not converge. So we use the boundary constraint to limit the representation locations to ensure training convergence as:

$$dis_{boundary} = \frac{1}{N} \sum_{i=1}^{N} ||D(r_i, \mathbf{0}) - constraint||_2 \tag{5}$$

where $N$ is the total number of samples and $r_i$ refers to the representation in a deep geometric space output from a certain linear layer of sample $i$. $D(r_i, \mathbf{0})$ is a distance metric of $r_i$ and origin of the space. $||\cdot||$ is $L2$ norm. $constraint$ is a parameter used to limit the distance of representations to the origin. $dis_{boundary}$ represents the average Euclidean distance of all samples to the boundary. When the number of categories is large, $dis_{boundary}$ can be set larger, so that different categories can find suitable spatial positions. However, if $dis_{boundary}$ is too large, the convergence will be slow. Minimizing $dis_{boundary}$ ensures that the representations will not infinitely far from the origin, then the training process can converge.

### 4.4 OVERALL OPTIMIZATION

The optimization of ASLoss is calculated as follows:

$$ASLoss = \lambda_{inner}dis_{inner} - \lambda_{outer}dis_{outer} + \lambda_{boundary}dis_{boundary} \tag{6}$$

where $\lambda_{inner}$, $\lambda_{outer}$ and $\lambda_{boundary}$ are three parameters. To ensure overall convergence, $\lambda_{boundary}$ should be slightly larger than $\lambda_{outer}$.

The final overall optimization of the whole network is computed as:

$$Loss_{overall} = \lambda_{asl}ASLoss + \lambda_{cls}Loss_{cls} \tag{7}$$

where $\lambda_{asl}$ and $\lambda_{cls}$ are two parameters and $Loss_{cls}$ refers to the classification loss such as cross entropy loss.

## 5 EXPERIMENT SETTINGS

### 5.1 TASK DESIGN, DATASETS AND EVALUATION METRICS

**Task Design and datasets**: We conduct experiments on two classification tasks that are extremely prone to confusion, i.e. *acne severity grading* and *facial expression recognition* to verify the effectiveness of the proposed method. Both tasks simultaneously have three confounding characteristics: data insufficiency, class imbalance, and vague evidence.

*Acne Severity Grading*: We use the ACNE04 dataset Wu et al. (2019) for acne severity grading. It is an open image dataset for facial acne severity grading. This dataset was collected from outpatients, so its number of class samples can reflect the true distribution of the severity of acne patients. The dataset totally contains 1457 images. According to the annotations of the ACNE04 dataset, there are four levels (four categories): mild, moderate, severe and very severe. For each category, 80% of

the samples are randomly divided into the training set and the rest are the test set. The class sample distribution is shown in Figure 3. First, the data is insufficient for deep learning where only 633 samples in the fattest class. Second, the class imbalance is also serious. The sample size of the largest class is almost five times than that of the smallest class. Next, the evidence for distinguishing among the four categories is unclear. The severity depends on the number of very fine-grained lesions on the whole face where 1 to 5 is mild, 6 to 20 is moderate, 21 to 50 is severe, and more than 50 is very severe. For models, inputs are large faces but the classification evidence is the number of extremely small objects which is very difficult to mine. This blurs the lines among the different categories. Some examples can be seen in Figure 4.

*Facial Expression Recognition:* We use a subset of the CelebA dataset Liu et al. (2015) for this task. CelebA is an open dataset containing 202,599 face images of 10,177 celebrity identities. Its annotations contain 40 attributes, such as eyebrows, eyes, nose shape, and more. We selected two attributes to form four categories: NO&NS (no mouth open and no smile), NO&S (no mouth open and smile), O&NS (mouth open and no smile), and O&S (mouth open and smile). Then, we randomly selected 1000 samples according to the class distribution of the ACNE04 dataset and divided 80% as the trainset and the rest as the test set as shown in Figure 3. Similarly, this subset is also insufficient, imbalanced and the evidence for distinguishing the four classes is also unclear because differences among categories are not significant. The inputs are similar faces but the identification should be focused only on the mouth.

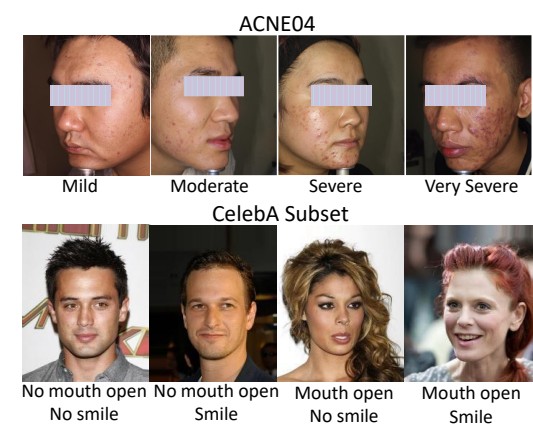

Figure 4: Examples for each category of the ACNE04 and CelebA subset.

**Evaluation Metrics**: We use the accuracy, precision, recall and f1-score commonly used in classification tasks as the evaluation metrics.

**Implementation Details** All models in the experiment use the same implementation settings. For the parameters of the ASLoss, the $\lambda_{inner}$ and $\lambda_{outer}$ are both set as 0.001 and the $\lambda_{boundary}$ is set as 0.01. For the parameters of the overall optimization, the $\lambda_{asl}$ and the $\lambda_{cls}$ are both set as 0.5. For training different networks, the learning rate is set as 1e-3, the batch size is set as 16 and the maximum training epoch is 400. Adam is chosen as the optimizer employed for training all models. All programs are coded using Python with PyTorch. All the experiments are conducted on a Linux Ubuntu with 32G RAM and an NVIDIA Geforce RTX 3090 GPU of 24G VRAM.

## 6 EXPERIMENT RESULTS, ANALYSIS AND DISCUSSION

### 6.1 EFFECTIVE OF AGGREGATION SEPARATION LOSS

To verify the effectiveness of ASLoss, we use ASLoss with different distance metrics on several mainstream deep vision networks. The mainstream vision networks include convolution-based VG-GNet Simonyan & Zisserman (2015), ResNet He et al. (2016), EfficientNet Tan & Le (2019) and transformer-based ViT Dosovitskiy et al. (2021), Swin-Transformer Liu et al. (2021), PVT Wang et al. (2021a) and T2T-Vit Yuan et al. (2021). For each network, we add linear mapping layers to the output backend of the original network to gradually reduce the number of features into two dimensions and then add a classification layer. The schematic diagram is shown in Figure 2. Next, we use ASLoss of three distance measures, L1 distance, L2 distance and cosine similarity, respectively, on the last two-dimensional feature layer. The comparison results on the two tasks of acne severity grading and facial expression recognition are shown in Table 1.

The performances of all the networks have been improved after using ASLoss, no matter the convolution-based or transformer-based network. For the severity grading task, the ASLoss with L1 distance metric improves the VGG16 network the most. The ASLoss with L2 distance brings the most improvement to the RES50, EFFICIENT_B3, Swin-Transformer and PVT networks. The ASLoss with cosine similarity improves the ViT and T2T-ViT the most significantly. Among all the

Table 1: Comparison results of mainstream deep vision methods not using and using ASLoss of different distance metrics on acne severity grading (left) and facial expression recognition (right). ACC: Accuracy, PRE: Precision, REC: Recall and F1: F1-score. L1: L1 distance, L2: L2 distance and COS: Cosine Similarity. ASL: ASLoss.

| Tasks | Acne Severity Grading | | | | Facial Expression Recognition | | | |
| Methods | ACC↑ | PRE↑ | REC↑ | F1↑ | ACC↑ | PRE↑ | REC↑ | F1↑ |
|---|---|---|---|---|---|---|---|---|
| VGG16 | 85.62 | 85.98 | 84.00 | 84.69 | 80.81 | 79.64 | 80.10 | 79.86 |
| VGG16+ASL(L1) | 85.96 | **86.73** | **85.27** | **85.59** | **82.83** | **82.59** | **82.97** | **82.46** |
| VGG16+ASL(L2) | **86.64** | 86.56 | 84.49 | 85.04 | 82.83 | 80.03 | 79.66 | 79.65 |
| VGG16+ASL(COS) | 85.96 | 84.54 | 82.71 | 83.30 | 82.83 | 81.82 | 79.29 | 80.05 |
| RES50 | 83.22 | 82.88 | 81.26 | 81.76 | 82.32 | 78.80 | 78.44 | 78.55 |
| RES50+ASL(L1) | 84.25 | 84.98 | 81.19 | 82.86 | 83.84 | 82.85 | **85.38** | **83.93** |
| RES50+ASL(L2) | **86.64** | **86.56** | **84.49** | **85.04** | **85.35** | **85.00** | 82.40 | 83.59 |
| RES50+ASL(COS) | 83.90 | 83.56 | 82.64 | 82.27 | 83.84 | 81.33 | 81.16 | 81.05 |
| EFFICIENT_B3 | 82.53 | 83.60 | 80.74 | 81.80 | 79.80 | 78.81 | 78.74 | 78.77 |
| EFFICIENT_B3+ASL(L1) | 86.30 | 86.19 | 83.90 | 84.88 | **83.84** | **83.50** | **84.13** | **83.73** |
| EFFICIENT_B3+ASL(L2) | **87.33** | **89.17** | **88.46** | **88.78** | 83.33 | 82.78 | 83.78 | 83.10 |
| EFFICIENT_B3+ASL(COS) | 85.62 | 86.40 | 85.08 | 85.44 | 83.33 | 80.88 | 81.81 | 81.30 |
| ViT | 65.75 | 66.93 | 64.94 | 65.44 | 66.67 | 57.30 | 47.30 | 47.57 |
| ViT+ASL(L1) | **82.88** | 83.04 | 81.19 | 81.09 | 82.32 | **81.59** | 80.32 | 80.90 |
| ViT+ASL(L2) | 82.53 | 82.63 | 79.42 | 80.12 | **83.84** | 81.37 | 79.51 | 80.31 |
| ViT+ASL(COS) | 82.53 | **83.75** | **82.99** | **83.29** | 81.82 | 81.17 | **82.74** | **81.01** |
| Swin-Transformer | 84.93 | 85.20 | 82.61 | 83.04 | 77.78 | 76.54 | 74.77 | 74.95 |
| Swin-Transformer+ASL(L1) | 85.96 | **85.92** | 82.84 | 83.83 | 80.30 | 75.95 | 79.43 | 77.20 |
| Swin-Transformer+ASL(L2) | **86.99** | 85.08 | **85.06** | **84.90** | **84.34** | **81.49** | **82.25** | **81.57** |
| Swin-Transformer+ASL(COS) | 83.22 | 82.81 | 78.11 | 80.14 | 81.31 | 80.62 | 76.10 | 78.08 |
| PVT | 80.48 | 80.71 | 78.83 | 79.71 | 75.25 | 72.20 | 68.80 | 70.10 |
| PVT+ASL(L1) | 83.90 | 82.04 | 81.34 | 81.46 | 76.26 | 78.67 | 70.57 | 73.65 |
| PVT+ASL(L2) | **83.90** | **82.35** | **81.34** | **81.66** | 77.78 | 79.06 | **73.73** | **75.99** |
| PVT+ASL(COS) | 83.56 | 82.38 | 79.84 | 80.80 | **79.29** | **79.39** | 71.63 | 74.48 |
| T2T-ViT | 82.53 | 82.14 | 79.64 | 79.75 | 79.80 | 78.97 | 76.10 | 76.83 |
| T2T-ViT+ASL(L1) | 83.22 | 81.67 | 82.26 | 81.83 | 78.79 | 81.10 | **76.74** | 78.68 |
| T2T-ViT+ASL(L2) | 83.90 | **84.74** | 83.19 | 83.67 | **81.82** | **85.25** | 76.62 | **79.86** |
| T2T-ViT+ASL(COS) | **84.25** | 83.42 | **84.79** | **84.08** | 80.30 | 79.13 | 76.32 | 77.25 |

networks, ASLoss brings the most obvious improvement to ViT with almost all evaluation metrics improved by about 15%. Meanwhile, the best performing model is EFFICIENT_B3 using ASLoss of L2 distance, with all evaluation results over 87%, surpassing the performances of all the other networks. For the facial expression recognition task, the ASLoss with L1 distance improves the VGG16 and EFFICIENT_B3 mostly. The ASLoss with L2 distance improves the Res50, Swin-Transformer and T2T-ViT most significantly. The ASLoss with cosine similarity brings the most obvious improvement to ViT and PVT. Among them, ASLoss brings the most obvious improvement to ViT, some evaluation metrics are even improved by about 20%. The best performing model is RES50 using ASLoss with L2 distance, with all evaluation results around 83%, surpassing most of the other network performances. Overall, ASLoss with L2 distance brings significant improvements on most networks.

First, the performance of these networks on the acne severity grading is slightly better than that on the facial expression recognition. On the one hand, the data volume for the acne severity grading is slightly larger than that of the facial expression recognition. On the other hand, mouth opening and smiling are more easily confused, compared to the number of small lesions. So the networks are more easily to obtain effective generalization on the acne severity grading task. Next, the transformer-based ViT-series networks do not perform significantly better than convolution-based networks. We believe that it is difficult for the ViT series of methods to converge at a good enough performance when the data is not sufficient for both tasks. In addition, the distance metrics that improve significantly on most networks are the L1 and L2 distances, although the ASLoss of all three distance metrics can improve the performance of different networks. In fact, when using L1

Table 2: Comparison results of using ASLoss on multiple linear layers in front of the classification layer on acne severity grading (left) and facial expression recognition (right). ACC: Accuracy, PRE: Precision, REC: Recall and F1: F1-score. L2: L2 distance. ASL: ASLoss.

| Tasks
Base Models
Layers | Acne Severity Grading
EFFICIENT_B3 + ASL(L2) | | | | Facial Expression Recognition
Res50 + ASL(L2) | | | |
|---|---|---|---|---|---|---|---|---|
| | ACC↑ | PRE↑ | REC↑ | F1↑ | ACC↑ | PRE↑ | REC↑ | F1↑ |
| Base model | 82.53 | 83.60 | 80.74 | 81.80 | 82.32 | 78.80 | 78.44 | 78.55 |
| ASLoss on the last one layer | **87.33** | **89.17** | **88.46** | **88.78** | 85.35 | **85.00** | **82.40** | **83.59** |
| ASLoss on the last two layers | 85.96 | 85.57 | 86.37 | 85.95 | 84.34 | 81.78 | 81.71 | 81.46 |
| ASLoss on the last three layers | 85.27 | 85.74 | 85.74 | 85.54 | **86.36** | 84.32 | 81.05 | 82.41 |
| ASLoss on the last four layers | 85.62 | 86.01 | 84.86 | 85.39 | 84.34 | 81.78 | 80.20 | 80.88 |

and L2 distances, ASLoss aggregates and diverges the representation locations of different classes in a deep space. While using cosine similarity, it gathers and diverges the angles of representation vectors. We believe that angular differences are still difficult to distinguish near the origin, compared to their positional differences in space. Therefore, ASLoss using L1 and L2 distance metrics performs slightly better than cosine similarity.

Such experimental results show that ASLoss can improve the performance of various mainstream visual networks, including convolution-based networks and transformer-based series. And the two best performing models, EFFICIENTNET_B3 and Res50, using the ASLoss of the L2 distance metric on the two tasks are selected to conduct the following experiments.

## 6.2 AGGREGATION SEPARATION LOSS ON MULTIPLE SPACES

we apply ASLoss to the last layer, the last two layers, the last three layers and the last four layers before the output layer, respectively, and compare their performances for both the acne severity grading task and facial expression recognition task. The schematic is shown in Figure 2. Since the best performing models on acne severity grading and facial expression recognition tasks are EFFICIENT_B3 and Res50 optimized by ASLoss of L2 distance metric, we continue to use these two networks as base models. The comparison results are displayed in Table 2. The first row in the table shows the results of the base model without ASLoss. Rows two to five show the experimental results of gradually increasing the number of representation spaces using ASLoss.

From the experimental results, the performance of the base model can be improved by using ASLoss on representations of all four multi-spatial strategies. For the acne severity grading task, the employment of ASLoss on four multi-space representations improves the accuracy of the base model by about 3%, the precision by about 2%, the recall by more than 4%, and the f1-score by more than 4%. Among them, the model optimized by ASLoss only on the one previous layer of the output layer achieves the best performance. Compared with the base model, the accuracy is increased by nearly 5%, the precision is increased by more than 5%, the recall rate is increased by nearly 8%, and the f1-score is increased by nearly 7%. For the facial expression recognition task, the results are similar to the acne severity grading. The employment of ASLoss on four multi-space representations improves the accuracy of the base model by over 3%, the precision by about 3%, the recall by around 3%, and the f1-score by about 4%. Among them, the model optimized by ASLoss only on the one previous layer of the output layer achieves the best performance. Compared with the base model, the accuracy is increased by over 3%, the precision is increased by more than 6%, the recall rate is increased by nearly 4%, and the f1-score is increased by over 5%.

For the performance on the two tasks, increasing the feature space employed by the ASLoss does not gradually improve the model performance. We believe that excellent performance is based on extracting enough representations. Premature aggregation and separation of feature representations may lead to insufficient feature extraction. At the same time, too many layers to perform feature aggregation and separation at the same time may lead to contradictions in gradient update, resulting in inconsistent update directions of network weights. Weight changes in the previous layers will cause the subsequent layers to change. Convergence will become difficult. Therefore, only the aggregation and separation of representations in the one layer before the classification layer achieves the best results.

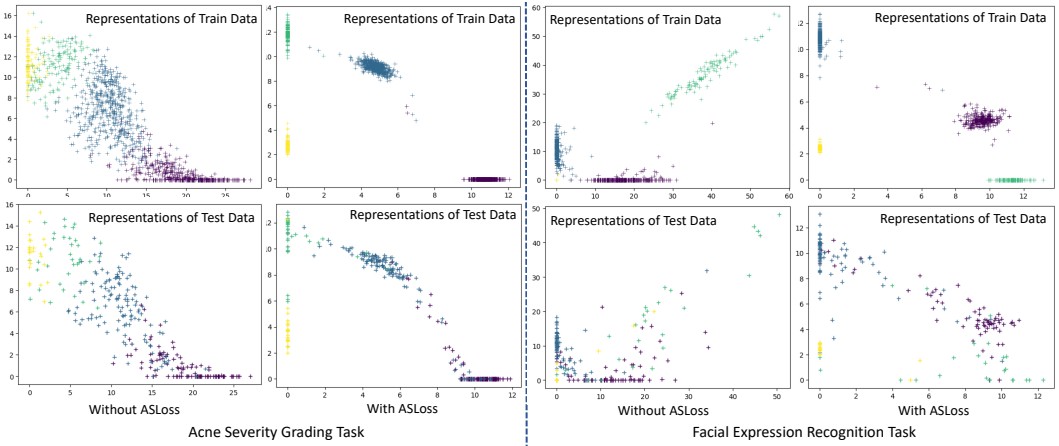

Figure 5: Visualization of representations. The left and right halves are for the acne severity grading (EF-FICIENT_B3 with ASLoss of L2 distance) and the facial expression recognition (Res50 with ASLoss of L2 distance). For each task, the left and right two subfigures are the representations extracted without and with ASLoss. The upper and lower subfigures are representations of train data and test data.

## 6.3 REPRESENTATION VISUALIZATION

We connect the representation extraction layer after the original network output and set the feature extraction layer finally to be two-dimensional. The two-dimensional output representations are visualized and compared as shown in Figure 5. The left and right halves are for the acne severity grading and the facial expression recognition tasks. The EFFICIENT_B3 and Res50 are selected for the two tasks to conduct this experiment due to their excellent performances.

For the acne severity grading task, the training sample representations extracted by the model without ASLoss have no obvious boundary and the distribution is scattered. For the training representations using ASLoss, the representations of the same category are more concentrated, and the samples of different categories can be clearly separated. For test data, the representation overlap is serious when not using ASLoss. After using ASLoss, the overlap of representations in space is alleviated. For the facial expression recognition task, three categories concentrated in the coordinate origin and coordinate axes in the training samples without ASLoss, leading to heavy overlap. After using ASLoss, the representations of different classes are spread out. For the test data representations, heavy overlap happens without ASLoss. After using ASLoss, the overlap is alleviated. All representations using ASLoss are constrained to the ideal range without disorderly dilation.

This result shows that ASLoss can effectively aggregate in-class samples to separate out-of-class samples. For the test samples of the two tasks, the effect of widening the distance of different classes is significant. This increases the distinguishability of these samples that are located among different class distributions. So the confusion among different classes can be clarified.

## 6.4 COMPARISON OF CONFUSION

The confusion matrixes for the two tasks are shown in Figure 6. The EFFICIENT_B3 and Res50 are also selected as the comparison models for the two tasks respectively. For each task, the left and right two confusion matrixes are from the models without and with ASLoss of L2 distance. The upper and lower confusion matrixes are for train and test data.

For both tasks, the use of ASLoss can improve performance of the models on not only the training data but also the test data. On the training data, both base models perform a little confusion on different categories. With the help of ASLoss, the confusion on the training dataset almost completely disappeared. On the test data, the predictions of the models without ASLoss perform confusion seriously. After using ASLoss on the models, the performance of the two classes 2 and 3 with insufficient data is improved sufficiently, while the performance of the two classes 0 and 1 with more samples decreases slightly. The overall accuracy is significantly improved.

This result demonstrates that the ASLoss effectively clarifies the confusion. Such results can be explained from visualization. The representation distances of the categories are pulled apart. The distribution of the same class is concentrated. So the confusion can be clarified.

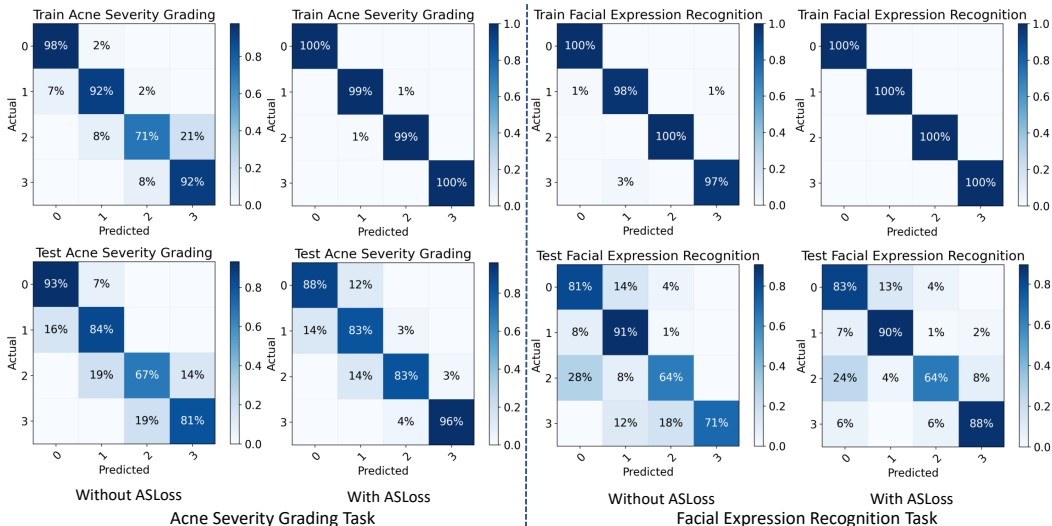

Figure 6: Comparison of confusion matrixes. The left and right halves are for the acne severity grading (EFFICIENT_B3 with ASLoss of L2 distance) and the facial expression recognition (Res50 with ASLoss of L2 distance). For each task, the left and right two confusion matrixes are from the models without and with ASLoss. The upper and lower confusion matrixes are for train data and test data.

Table 3: Comparative experimental results with state-of-the-art methods on acne severity grading (left) and facial expression recognition (right). ACC: Accuracy, PRE: Precision, REC: Recall and F1: F1-score.

| Tasks | Acne Severity Grading | | | | Facial Expression Recognition | | | |
| Methods | ACC↑ | PRE↑ | REC↑ | F1↑ | ACC↑ | PRE↑ | REC↑ | F1↑ |
|---|---|---|---|---|---|---|---|---|
| BaseModel + Focal Loss | 82.19 | 81.75 | 79.26 | 79.25 | 84.85 | 83.32 | **82.62** | 82.94 |
| BaseModel + Weighted CE | 83.90 | 83.87 | 81.70 | 82.38 | 83.84 | 81.38 | 82.31 | 81.69 |
| BaseModel + CBLoss | 82.88 | 81.94 | 79.32 | 80.46 | 82.83 | 80.57 | 79.77 | 80.13 |
| BaseModel + TDE | 85.27 | 85.32 | 83.22 | 83.78 | 84.85 | 82.91 | 81.74 | 82.15 |
| BaseModel + ASLoss(ours) | **87.33** | **89.17** | **88.46** | **88.78** | **85.35** | **85.00** | 82.40 | **83.59** |

## 6.5 Comparison with State-of-the-Arts

We compare the proposed ASLoss with some state-of-the-art methods including Focal Loss Lin et al. (2020), Weighted Cross Entropy Wang et al. (2017b), CBLoss Cui et al. (2019) and TDE Tang et al. (2020) as shown in Table 3.

From the experiment results, the proposed ASLoss outperforms other methods on all metrics for the acne severity grading task. For the facial expression recognition, the overall performance of ASLoss outperforms all other methods, although the recall of Focal Loss is slightly higher than the ASLoss. The experimental results show that the performance of ASLoss reaches the state-of-the-art level.

## 7 Conclusion

In this paper, we propose a novel, simple and intuitive ASLoss, as an adjunct for classification loss to clarify the confusion in some common cases. The ASLoss aggregates the representations of the same class samples as near as possible and separates the representations of different classes as far as possible to mine the commonalities of the same class and the gaps among different classes. We use two classification tasks with three simultaneous common confounding characteristics i.e. data insufficiency, class imbalance, and unclear class evidence to demonstrate the performance of ASLoss. We conduct representation visualization, confusion comparison, and detailed comparison experiments. The experimental results show that the model using ASLoss can extract sufficiently clear and distinguishable representations in deep spaces, the confusion among different classes is significantly clarified and the best-performing network reaches the state-of-the-art level.

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
