# OpenReview forum: "Effectively Clarify Confusion via Visualized Aggregation and Separation of Deep Representation"
_ICLR.cc/2023/Conference — Submitted to ICLR 2023_

### Official Review · Reviewer_xFz5 · 2022-10-24

**Confidence:** 4
**Clarity, Quality, Novelty And Reproducibility:** I found the paper lack of novelty. Th…
**Correctness:** 3
**Technical Novelty And Significance:** 1
**Empirical Novelty And Significance:** 1
**Recommendation:** 3

**Details Of Ethics Concerns:**

No ethics concerns.

**Strength And Weaknesses:**

Pros:
The authors implement the ASLoss with different model architectures and show the effectiveness of the method.

Cons:
The idea is exactly the same as metric learning loss. The three components, inner aggregation, outer seperation and boundary constraints are widely used in well-known paper like SimCLR and other contrastive learning paper or triplet loss paper. I do not see any references related to metric learning and the experiments in the paper are not strong enough to support that this method is very effective in the imbalanced learning or few-shot learning scenarios. To insert the ASLoss into classification problem is similar to the idea of supervised contrastive learning. I do not see enough novelty in this paper.

Some other baseline models should be considered in imbalanced learning such as LDAM.

**Summary Of The Paper:**

The paper presents a loss function called ASLoss as an adjunct for classification loss to address the confusion issue under data insufficiency and class imbalance cases. The ASLoss can be decomposed into inner aggregation, outer separation and boundary constraint, which contraints the representation to have some invariant characteristics. The effectiveness is demonstrated by experiments.


**Summary Of The Review:**

The authors ignored the work in metric learning e.g., contrastive learning and triplet loss, and proposed a loss function ASLoss which is very similar to those paper. I found the paper lack of novelty and the experimental results are not strong enough to support that this loss function is critic to this setup.

---

> ### Author Response · Authors · 2022-11-16
> **Reply**
>
> Thank you very much for your careful reading and valuable comments.
>
> 1. The idea is exactly the same as metric learning loss. The three components, inner aggregation, outer seperation and boundary constraints are widely used in well-known paper like SimCLR and other contrastive learning paper or triplet loss paper.
> To insert the ASLoss into classification problem is similar to the idea of supervised contrastive learning. I do not see enough novelty in this paper.
> I do not see any references related to metric learning.
>
> Thank you for your comments. You are right.
>  Indeed contrastive learning including simCLR uses an agent task to improve the classification.
> The agent task transforms the samples with different strategies.
> While our method does not need to transform the samples but clarify them in deep spaces.
> We will add more discussions about this in our paper to detail the differences. Thank you.
>
> 2. the experiments in the paper are not strong enough to support that this method is very effective in the imbalanced learning or few-shot learning scenarios.
>
> Indeed, we propose that some cases are easily confused: similar samples, insufficient data, imbalance class, and unclear class boundaries but not limited only to imbalance.
> As for the few shot learning, it is only one aspect of insufficient data, but also too much insufficient which is not common.
> So we did not compare shot methods in this paper.
>
>
>
>
> 3. Some other baseline models should be considered in imbalanced learning such as LDAM.
>
> Thank you for your suggestion. It is our negligence that led to inadequate research.
> We will include a comparison of this approach in the final release.
>
>
> Hope you reconsider our paper.
> No matter what the result is, we thank you for your careful reading.

---

### Official Review · Reviewer_7v3V · 2022-10-24

**Confidence:** 4
**Correctness:** 2
**Technical Novelty And Significance:** 2
**Empirical Novelty And Significance:** 1
**Recommendation:** 3

**Clarity, Quality, Novelty And Reproducibility:**

The clarity of this paper is rather well. The quality and novelty of this paper are limited. Please see the section "Weaknesses" for details. For the empirical results, it seems rather well to reproduce them.


**Strength And Weaknesses:**

## Strength
1. Overall, this paper is written rather clearly.
2. The considered problem is relevant to the community.


## Weaknesses
1. The "confusion" issue, as the authors claimed,  lacks a formal definition and seems a little confusing.
2. This paper lacks theoretical results to support the claims.
3. For the proposed ASLoss, the novelty is limited since the ideas of aggregating intra-class samples and separating inter-class samples are widely used in image classification. Besides, some parts of the notations are confusing and wrong. Specifically, in Eq.(1), the notation $N_{c_i = c_j}$  is confusing since what $i$ and $j$ come from? The similar issue also occurs for the notation $N_{c_i \neq c_j}$ in Eq.(2).
4. For the experiments, the hyper-parameters of the proposed method are selected on the test datasets during training and should be searched by another validation set. Moreover, this paper lacks ablation studies about the different parts of the proposed loss function.

**Summary Of The Paper:**

This paper proposes a new loss function, e.g., Aggregation Separation Loss (ASLoss), to clarify confusion to improve image classification performance.
Specifically, the ASLoss aggregates the representations of the same class samples as near as possible and separates the representations of different classes as far
as possible. Experimental results on two datasets (in the case of data insufficiency, class imbalance, and unclear class evidence) are conducted to illustrate the effectiveness of the proposed method.

**Summary Of The Review:**

Overall, I think this paper proposes a new loss function for image classification where the novelty is limited and its effectiveness lacks theoretical support and ablation studies. Thus, the quality is poor and I suggest rejection.

---

> ### Author Response · Authors · 2022-11-16
> **Reply**
>
> First of all, thanks for your careful reading and valuable comments.
>
> 1. The "confusion" issue, as the authors claimed, lacks a formal definition and seems a little confusing.
>
> Very reasonable, we will add this part to the article for a more detailed definition and explanation.
>
> For any sample $x \in \mathbf{X}$ where $\mathbf{X}$ is a category, if a person or a model considers $x$ to belong to another category $\mathbf{Y}$, this misperception is confusion.
> Confusion often occurs when the samples are very similar, the perceiver is inexperienced, and the class boundaries are not clear.
> For an example of similar samples, people often misidentify twins.
> For an example of inexperience, children who have never seen a tiger will consider a young tiger as a cat.
> The insufficient data and imbalanced class in machine learning belong to this case.
> For an example of the unclear boundary, it is difficult to determine whether a person is middle-aged or young based on only appearance.
>
>
> 2. This paper lacks theoretical results to support the claims.
>
> Indeed, as you say, we propose this work not on the basis of some theory, but on the basis of intuition.
> we propose that some cases are easily confused: similar samples, insufficient data, imbalance class and unclear class boundaries.
> We think that when humans distinguish between categories, they search in their cognition for a category that satisfies sample characteristics.
> And it would be reasonable to reinforce the heterogeneous boundaries and the homogeneous commonalities in cognition. Therefore, we proposed the mutual promotion of the three parts of ASLoss.
>
>
> 3. For the proposed ASLoss, the novelty is limited since the ideas of aggregating intra-class samples and separating inter-class samples are widely used in image classification.
>
> Thank you for your question. You make a lot of sense.
> There are various methods to optimize distance for different tasks, but they are more or less limited.
> Such as CosFace, maximum the cosine distance between different person faces.
> But the same face is similar enough compared to other person faces.
> While our method is also suitable for one class containing various samples.
> We will also discuss your questions further in the article.
>
>
>
>
>
>
> 4. Besides, some parts of the notations are confusing and wrong
>
> Thank you for pointing them out.  We will revise our paper correctly.
>
>
>
>
> 5. For the experiments, the hyper-parameters of the proposed method are selected on the test datasets during training and should be searched by another validation set. Moreover, this paper lacks ablation studies about the different parts of the proposed loss function.
>
> We apologize for this oversight. Thank you for your point out.
> In fact, the description in the article is wrong. We selected the model using the validation set, which is included in the training set. The test set was not used during the training. We will revise that part of the article.
>
> For the ablation study, the ASLoss is not an incremental combination. The three parts work together.
> One part pulls, one part pushes, and one part limits the scope.
> If deleting any one of them, this method will not work well.
> If deleting pulling, the same class samples will divergence distributed.
> If deleting pushing, different classes will be near to each other.
> If deleting the scope limitation, the loss will not converge.
>
>
>
> All in all, hope you reconsider our paper.
> No matter the results, thank you for your careful reading and comments.

---

### Official Review · Reviewer_nFtj · 2022-10-24

**Confidence:** 3
**Correctness:** 2
**Technical Novelty And Significance:** 2
**Empirical Novelty And Significance:** 2
**Recommendation:** 3

**Clarity, Quality, Novelty And Reproducibility:**

Clarity is good, the paper is easy to follow and clear to read in most places.
Quality and novelty are not good enough: the loss function seems not new enough. Quality needs to be improved.
Reproducibility is a concern: no directly comparison with prior works, datasets are too limited.


**Strength And Weaknesses:**

The biggest strength of this paper is clarity, the idea and motivation is clearly stated and verified through experiments.

The biggest concerns are:
1. Novelty: the idea seems to be not new. For instance in CosFace(https://arxiv.org/pdf/1801.09414.pdf) the author proposed to maximize the margin of Cosine distance.
2. Experiments: the authors claim that it will benefit the models in extreme cases such as: data insufficient, class imbalance etc. But they only experimented two (unpopular) datasets: Acne Severity gradient and CelebA. Clearly, more experimental supports are needed.
3. Unable to calibrate this method with other baselines. Due to unpopular datasets, it is almost impossible to directly compare this method with baselines. Although authors compare quite a few baselines in Table 3, those are not official numbers. For instance in CBLoss(Cui et al.), Long-Tailed CIFAR, iNaturalist, ImageNet are studied. But in this paper the author ignored all those datasets. Because of this, we are not able to make fair comparisons.
4. (Minor) The formatting of this paper can be improved, many figures/plots are in low res.

**Summary Of The Paper:**

This paper proposed a new training loss that they claim to be helpful in data insufficient, class imbalance and unclear class evidence scenarios. The core idea is the newly created loss called ASLoss: it tries to maximize the similarities of samples belonging to the same classes; while minimize the similarities of samples across the classes. Despite the simple idea, the authors are able to obtain good results in Acne Severity Grading dataset and CelebA dataset.

**Summary Of The Review:**

The main reasons I tend to reject are listed in the previous sections.

---

> ### Author Response · Authors · 2022-11-16
> **Reply**
>
> Thank you for your careful reading and thoughtful comments.
>
> 1. Novelty: the idea seems to be not new. For instance in CosFace the author proposed to maximize the margin of Cosine distance.
>
> Thank you for pointing out our shortcomings. We will revise the article and discuss it.
> Their work maximizes the cosine similarity and introduces intervals. And our work is to optimize the clustering between samples, to maximize or minimize, and the distance is not limited to a certain kind of distance. And the intervals they work with is similar to what we do with maximizing class spacing. In fact, the two work from different starting points. They're on facial recognition, and we're clarifying confusion, which may be more generalizable. The three parts of our ASloss are combined together to work but not only optimizing one of them.
>
>
>
> 2. Experiments: the authors claim that it will benefit the models in extreme cases such as: data insufficient, class imbalance etc. But they only experimented two (unpopular) datasets: Acne Severity gradient and CelebA. Clearly, more experimental supports are needed.Unable to calibrate this method with other baselines. Due to unpopular datasets, it is almost impossible to directly compare this method with baselines. Although authors compare quite a few baselines in Table 3, those are not official numbers. For instance in CBLoss(Cui et al.), Long-Tailed CIFAR, iNaturalist, ImageNet are studied. But in this paper the author ignored all those datasets. Because of this, we are not able to make fair comparisons.
>
>
> Thank you for the comment.
> We found that your main concern was two aspects:
>
> 1） we could not prove the confusing cases we presented using just two data sets.
>
> 2） the uncommonly used dataset can not compare these baselines fairly.
>
>
> 1）. we propose that some cases are easily confused: similar samples, insufficient data, imbalance class and unclear class boundaries.
> Although we use only two data sets, the two data sets satisfy all of these conditions.
>
> (1.1) acne severity grading:
> - data volume not large.
> -  class imbalance
> -  unclear boundary(classification evidence is based on the various small lesions on one face but not the whole large target)
> -  similar samples (the same face at different times may belong to different severity because of treat or worsen)
>
> (1.2) expression recognition:
> - data volume not large(we randomly select some of them).
> - class imbalance(we manually constrict the imbalance).
> - unclear boundary (we do not identify the face, but the expression (smile &mouth open | smile &mouth close | not smile | mouth open | not smile | mouth close)
> - similar samples(the same face may not have the same expression).
>
> It's true that, as you say, we don't use mainstream data sets. However, it is difficult for other data sets to satisfy these cases simultaneously. These cases are widely heard in medical problems, but medical data are difficult to obtain. So we constructed the data that fit best.
>
>
> 2） The main reason is also like the descriptions above. Other data sets could not meet the above cases, so these comparison methods did not have official scores and were reproduced by us. But you can think the comparison is fair. Because they are reproduced using the same configurations as our methods.
> In addition, few methods focus on the same key as us which clarifies confusion in many cases.
> That is to say, no sota methods for clarifying confusion but on special cases such as imbalance, insufficiency, or else.
> So we cautiously select some methods for special cases to compare.
> Hope you can understand our motivation.
>
>
> 3. (Minor) The formatting of this paper can be improved, many figures/plots are in low res.
>
> We will modify this issue. Thanks for your advice.

---

### Official Review · Reviewer_Ziee · 2022-10-26

**Confidence:** 4
**Clarity, Quality, Novelty And Reproducibility:** Overall, it is a good paper based on …
**Correctness:** 3
**Technical Novelty And Significance:** 2
**Empirical Novelty And Significance:** 2
**Recommendation:** 5

**Strength And Weaknesses:**

Quality/Clarity: the paper is well written and the techniques presented are easy to follow. Its motivation is to aggregate the representations of the same class samples as near as possible and separate the representations of different classes as far as possible. And the authors design the corresponding Aggregation Separation Loss (ASLoss) to learning these representations. The method is validated our method on two image classification tasks and show promising results.

Originality/significance: the idea is incremental, which combines multiple loss functions (most are known) together to clarify confusion and improve separation over different classes. Extensive experiments are conducted to show its advantage on confusion-caused common conditions: data insufficiency, class imbalance, and unclear evidence.

**Summary Of The Paper:**

This paper proposes a novel, simple and intuitive Aggregation Separation Loss (ASLoss), which aggregates the representations of the same class samples as near as possible and separates the representations of different classes as far as possible. The authors conduct extensive experiments on diffirent scenarios i.e. data insufficiency, class imbalance, and unclear class evidence to demonstrate ASLoss, and the results show that representations in deep spaces extracted by ASLoss improve classification performance and reaches the state-of-the-art level.

**Summary Of The Review:**

The idea is incremental, where ASLoss combines multiple loss functions (most are known) together to clarify confusion and improve separation over different classes.

---

> ### Author Response · Authors · 2022-11-16
> **Reply**
>
> Thank you for your question.
>
> We find that your main concern is the novelty of this article.
> You think our proposed approach is a combination of known methods.
>
> In fact, we did combine the proposed ASLoss into classification losses as an auxiliary. However, the three parts of ASLoss we proposed are not in the existing other works.
>
> Indeed, some approaches are similar to ours but different. We did not discuss the difference between them, which is our negligence.
> This part of the discussion has been added to the new version of the article.
> Our point of view focuses on abstracting and unifying various confusing cases, trying to clarify confusion in a higher dimension.
>
> Hope you can reconsider.
>
> No matter what the result is, we thank you for your careful reading and correct guidance.

---

### Decision · Program_Chairs · 2023-01-20

**Decision:**

Reject

**Justification For Why Not Higher Score:**

Limited novelty, lack of theoretical results, unconvincing experiments

**Justification For Why Not Lower Score:**

N/A

**Metareview: Summary, Strengths And Weaknesses:**

The paper proposes a new loss function designed to aggregate samples of the same class together and separate this of different classes. The reviewers point out that the idea is incremental, and I have to agree. In essence, all loss functions are designed to separate samples of different classes and the authors have not provided a theoretical justification or sufficient experiments to show that this procedure achieves something that's spectacularly different than prior deep learning architectures. The authors point out that data that fits all the challenging situations simultaneously is difficult to find/acquire.  In this case, the benefits of the method should be illustrated on data that only partially fulfills the criteria. If the situations when the method is best suited are extremely infrequent, that also makes the method less useful.


**Summary Of Ac-Reviewer Meeting:**

N/A